# Sex and rank in public service hierarchies: Rank distribution in Ghana's health and security services

Ishmael D. Norman[1,2], Emmanuel Dodzi Kpeglo[3]*, Saralees Nadarajah[4]

1 Academic Division, Ghana Armed Forces Command and Staff College, Accra, Ghana, 2 Department of Criminology, Institute for Security, Disaster and Emergency Studies, Langma, Ghana, 3 Department of Economics and Actuarial Science, University of Professional Studies, Accra, Ghana, 4 Department of Mathematics, University of Manchester, Manchester, United Kingdom

* emmanuel.kpeglo@upsamail.edu.gh

## Abstract

This study analyses leadership patterns in Ghana's health and security institutions since 1992, with a particular emphasis on the sex composition of senior positions in the Ghana Health Service, the Armed Forces, and the Police Service. A mixed-methods approach was employed, comprising a qualitative literature review, quantitative analysis of the Ghana Police Service rank hierarchies, assessment of the Military Occupational Physical Assessment Test in relation to Military Occupational Specialties, and content analysis of relevant sections of the Affirmative Action Act (2024). In 2025, women represented 39% of doctors (5,068/12,900), 30% of police officers (12,945/43,968), and 15% of soldiers (2,400/16,000) in Ghana. Leadership in the Ghana Armed Forces remained male-dominated, with fewer than five female Generals among 115 in the forces and only one female Inspector-General of Police since 1992. Statistical analysis of police rank distribution showed a significant association between sex and rank ($\chi^2$, $p < 0.05$), indicating persistent disparities in career progression. Findings highlight systemic institutional barriers affecting women's advancement in Ghana's health and security sectors. Targeted institutional reforms aligned with the requirements and merit-based principles of the Affirmative Action Act (2024) are necessary to address the disparities and strengthen equitable representation.

## 1. Introduction

### 1.1. The purpose of the Affirmative Action Act

Ghana passed its Affirmative Action Act in 2024 (Act 1121) to address sex-based inequities and promote human flourishing [1]. The Affirmative Action Act was necessitated by a prima facie case of sex-differentiated promotions across several public institutions, particularly at senior administrative and decision-making levels [2]. For

**Data availability statement:** An anonymised minimal dataset sufficient to reproduce all statistical analyses reported in this study (including Tables 2 and 5–7) has been deposited in a public repository (https://zenodo.org/records/17747718). The repository also contains the SPSS syntax file (analysis_syntax.sps) and the SPSS output file (analysis_output.spv) used to conduct the chi-square tests and the ordinal logistic regression.

**Funding:** The author(s) received no specific funding for this work.

**Competing interests:** The authors have declared that no competing interests exist.

example, as of 2025, the Ghana Armed Forces (GAF) had 115 top-brass officers, including Brigadier-Generals and Major-Generals; fewer than five were women [3].

There are many reasons why the military hierarchy in Ghana remains male-dominated, including structural, administrative, and organisational forms of bias. From a structural perspective, the design of recruit training batteries is not gender-neutral. The anatomy of men differs from that of women. It is therefore reasonable to expect different performance outcomes when both groups are subjected to uniform physical standards, such as identical repetitions of callisthenics or standardised endurance-run benchmarks. Such uniform standards may inadvertently disadvantage women and reinforce sex-based disparities in progression through the military ranks.

## 1.2. The role of civilian influences over the security and other critical agencies

Even without the Affirmative Action Act's imperative for the military, politically motivated "grade creep" continues, reflecting a recurring tendency to restructure commands in ways that justify rank elevations detached from operational requirements [3]. Some of the Military Occupational Specialties (MOS) that women are encouraged to select have a career ceiling at the rank of Colonel, thereby limiting their potential for advancement beyond that rank. This policy raises equity concerns, as ranks at Colonel and above in the GAF retire on their basic salaries and receive additional emoluments. As a result, the inability to advance beyond Colonel creates significant economic disparities between, for example, a Colonel and a Brigadier General.

Recent promotion patterns in the Ghana Army have disproportionately favoured men over women. As a result, implementing the Affirmative Action Act within the security services may place additional strain on existing personnel structures. Since current male officers benefitting from promotions, authority, and privileges cannot easily be displaced, the financial cost of achieving sex parity through affirmative action is expected to rise substantially. Such adjustments place pressure on defence budgets, potentially affecting the security services' capacity to acquire equipment, invest in infrastructure, provide adequate office resources and staff support, and meet increased operational and retirement-related expenditures [3].

In Festus Aboagye's 2025 policy brief *Counting the Stars: The Paradox of Sporadic Military Rank Inflation in Ghana's Army*, he noted that "African militaries often display a disproportionate ratio of officers to enlisted personnel, resulting in degrees of organisational inefficiencies" [3]. These dynamics suggest that both political influence and structural constraints interact to shape promotion outcomes, often to the detriment of women in the military hierarchy.

## 1.3. Gendered leadership of Ghana Armed Forces: A long history of exclusionary practices

The GAF's leadership has been exclusively male from 1961 to 2024, reflecting persistent sex-based biases and exclusionary practices established partly under British expatriate military leadership during the colonial era. However, this historical legacy does not fully account for the contemporary limitations on women's advancement.

Globally, military institutions, except for a few cases such as Israel, have historically restricted women from combat roles, assigning them primarily to support services. Over time, this approach shifted, and today most military units allow women to serve in a broader range of units, including some combat or ranger regiments.

The opportunity to serve in combat situations provides the soldiers with a broad set of competencies, including critical decision-making, logistics and supply chain management, and crew and equipment oversight. In Ghana, however, junior ranks were historically managed by mid-level officers or enlisted cadres, many of whom were women restricted to subordinate clerical, administrative, or purely supportive tasks, limiting their professional development relative to men [4].

The administration of power within the GAF was based on paternalistic and segregated organisational norms, which were similar to the segregation that was observed in the security services and recruitment of locals or non-white citizens during the colonial era in Ghana, Nigeria, and in South Africa under the Apartheid Regime between 1948 and 1994 [5,6]. The gender dynamics within the GAF before the promotion of Brigadier-General Constance Edjeani-Afenu, and later Brigadier-General Felicia Twum-Barima, can be characterised as those of a predominantly male military establishment shaped by entrenched patterns of inadvertent discrimination and gender segregation. These patterns operated through institutional directives, orders, and customs that functioned as de facto mechanisms of sex-based exclusion within the army.

Civilian leadership, especially at the executive level, often endorsed or perpetuated these systemic practices, which frequently curtailed women's advancement and representation at the highest decision-making levels of the military hierarchy.

The situation became even more challenging during Ghana's extended period of military involvement in national politics. From 1966 onwards, the GAF became deeply entangled in governance, pushing issues of gender equity and personnel development to the back burner, a dynamic that some scholars argue contributed to the conditions that enabled repeated coup d'états [4].

In 1961, the appointment of the first Ghanaian professional soldier as Chief of Defence Staff appeared to signal a new era, one that promised inclusive personnel development and improved civil-military relations, consistent with Samuel Huntington's notion of "objective civilian control," wherein military authority was intentionally curtailed to guarantee civilian dominance [7].

However, this shift was short-lived. President Kwame Nkrumah, who initiated the professionalisation reforms, soon prioritised consolidating authoritarian rule, thereby undermining the foundations of inclusive personnel development and disrupting the emergent balance in civil-military relations.

From independence in 1957 until the return to constitutional rule in 1992, Ghana's civil-military relations defied premise classification. It would be intellectually dishonest to claim that the country consistently adhered to either Huntington's "separation" model or Janowitz's "concordance" model. Instead, the historical evidence suggests alternating forms of "Subjective Military Control" by civilian administrations (from roughly 1920–1966) and "Subjective Civilian Control" by the military governments (from 1966 to 1992), punctuated by brief returns to constitutional governance.

The promising prospect of a stable, professional civil-military partnership that appeared to emerge around 1961 was disrupted by Nkrumah's rapid shift toward authoritarian consolidation, which weakened the institutional foundations for balanced, professionalised civil-military engagement.

In *The Soldier and the State*, Huntington analysed the interdependent relationship between civilian political authorities and the military, emphasising that civil-military relations form a "system composed of interdependent elements." These elements include the military's structural position in the government, the informal influence of military actors in politics and society, and the ideological orientations of both military and civilian institutions [8].

Huntington's framework distinguishes between "objective civilian controls," which promote professional autonomy for the armed forces alongside political neutrality, and "subjective civilian controls," which occur when political authorities deliberately shape or dominate military institutions to preserve their own power. He argued that the effectiveness and professional integrity of the officer corps depend on striking the right balance between these forms of control, thereby ensuring that national security is managed responsibly [9].

Researchers of Ghanaian defence have argued that Ghana aligns more closely with the Constitutional/Prescriptive Model rather than Huntington's Separation Theory or Schiff's Concordance Model. As stated in *Analysis of Civil-Military Relations of Ghana from 1957 to 2022* [9], Ghana's system grants the government enough authority over recruitment, promotions, and retention within the GAF. This dynamic operates as a form of constitutional encroachment by the central government on the military's internal affairs.

The implementation of the Affirmative Action Act adds another layer of external pressure on the internal human resources management in the Ghana Army and the broader public service. Because the central government is the primary sponsor and overseer of the Act, it is expected to scrutinise recruitment and promotion patterns, as well as personnel data, in the security services more closely. Although Ghana adheres to the Constitutional/Prescriptive Model of civil-military relations, this oversight inevitably shapes the institutional environment in which officers, especially women, are evaluated and promoted.

The attributes of the Constitutional/Prescriptive Model are that the state has the power to design, regulate, and oversee activities within civil-military relations. The state has intentionally, and as mandated by constitutional provisions, subordinated the armed forces under civilian authority [10–12]. There must be apparent acceptance and acquiescence to the subordination, supported by legislation that operationalises constitutional directives [11]. This model also requires demonstrable, accountable performance of security and military functions within the constitutional and legislative framework.

A deviation from the constitutional construct is impermissible and may constitute intervention or treason, particularly if initiated by the military. Although the Constitutional Model promotes collaboration and cooperation, it establishes clear expectations and behavioural boundaries for the civilian authorities and military elites. As a result, neither Separation Theory nor Concordance Theory fully characterises civil-military relations in Ghana. Instead, as argued by [13] and [9], Ghana's system is best understood as a subjective military control structure, shaped by constitutional mandates and civilian oversight.

It is essential to recognise that Huntington's formulations were developed in the context of the United States military establishment, which differs considerably from the GAF in structure, recruitment procedures, promotion systems, mission orientations, and MOSs. While the U.S. military enjoys substantial professional autonomy in both operational and administrative matters, the GAF operate under more constrained institutional independence. Autonomy within the GAF is primarily restricted to internal organisation and administration rather than executive or strategic decision-making, which remains firmly under civilian control.

In Ghana, the concept of autonomy is often understood and valued only by a relatively small segment of the population, typically those with higher levels of education or exposure. Due to a deeply ingrained paternalistic political culture, many citizens tend to defer unquestioningly to authority figures, whether they be military commanders, senior bureaucrats, or political leaders. This deference reinforces an environment in which both individual and institutional agency remain constrained by hierarchical expectations and assumptions that those in authority inherently "know best."

Autonomy is fundamental to objective civilian-military control, yet in Ghana it is limited by civilian authority over finances, procurement, and expenditure, creating persistent tensions between military professionalism and bureaucratic constraint [14]. Because budgetary power rests with the civilian Minister of Finance, even highly trained officers remain dependent on civilian decisions, even though they are expected to maintain political neutrality, thereby reflecting Feaver's (1996) question of "how to reconcile a military strong enough to act when civilians demand it, yet subordinate enough to act only when civilians authorise." Consequently, the professional officer becomes a bureaucratic agent of the state, entrusted with managing the legitimate use of force, and must uphold competence, ethical responsibility, and institutional loyalty, distinguishing the officer corps from enlisted personnel [15,16].

Rebecca Schiff's Concordance Theory emphasises alignment among the military, civil society, and political leadership to promote stable civil-military relations, treating these actors as "partners" [17]. Stability, she argues, depends on agreement across four indicators: *the social composition of the officer corps; the political decision-making process; the recruitment method; and the military style*. When these align, the likelihood of military intervention is reduced.

However, evidence from the *Analysis of Civil-Military Relations of Ghana from 1957 to 2022* [9] shows that these indicators have not consistently existed in the GAF or many Common Law African militaries [18]. Thus, the mere absence of coups does not prove concordance, especially when deeper misalignments in recruitment, culture, and decision-making persist. Because these indicators are often weak or absent, the intervention-reducing effect of Schiff's model is difficult to measure. Her framework assumes self-correcting institutions, yet in Ghana and similar countries, constitutional constraints on the military remain fixed unless altered by a coup, insurrection, or formal review. It is worth noting that neither [18] nor [8] addressed the longstanding gender exclusion in military institutions.

According to the investigation conducted by Colonel Festus Aboagye (Rtd.), Ghana's military has experienced a marked expansion in its General Officer positions, from a relatively stable 56 positions in 2004–125 in 2024, despite only a modest increase in total from just over 20,000 to under 30,000 personnel [3]. Despite the growth in high-ranking positions, women remain significantly underrepresented, suggesting the persistence of selective institutional barriers within the military.

He further observed that recent developments may reflect an implicit bias against women officers, whose ranks appear not to have been evaluated or promoted on the same terms as their male counterparts. This bias is demonstrated by their noticeable absence from recent promotion lists, even as the practice of "grade creep," a politically influenced rank inflation, continues, largely benefitting male officers.

Colonel Festus Aboagye (Rtd.) also documented a notable pattern of rank inflation beginning in 2000 with the establishment of regional commands. This restructuring resulted in widespread rank elevations within the Ghana Army, including the promotion of General Officers Commanding (GOCs) from Brigadier to Major General, among them Brigadier General Francis Yahaya Mahama, Commander 1 Infantry Brigade/Southern Command, and Brigadier General Henry K. Anyidoho, Commander 2 Infantry Brigade/Northern Command, with equivalent increases in Naval and Air Force ranks.

Most notably, Flight Lieutenant Rawlings promoted all service chiefs from Major General to Lieutenant General (and equivalent ranks) before leaving office: Major General Joseph Henry Smith to Lieutenant General, Rear Admiral E.O. Owusu-Ansah to Vice Admiral, and Air Vice Marshal John Asamoah Bruce to Air Marshal. These officers served alongside Lieutenant General Ben K. Akafia, who was CDS. Brigadier General Charles A. Okae, Chief of Staff at the General Headquarters (GHQ), was promoted to a two-star rank.

However, these promotions, particularly at the GHQ, were commonly perceived as politically motivated rather than merit-based, prompting President John Agyekum Kufuor to reverse many of them in March 2001 upon taking office" [3]. This episode illustrates how political considerations can directly shape structures, often to the detriment of transparent, merit-based promotion pathways, particularly for women.

As of today, GAF has fewer than ten women holding the ranks of Brigadier-General or Major-General combined, a situation that presents a prima facie case of historical, institutional, and systemic exclusion of women in critical decision-making roles within the Ministry of Defence. Elements of Ghana's military culture can be interpreted through frameworks such as Samuel Huntington's (1957) Separation Theory, Rebecca Schiff's (2008) Concordance Model, and related civil-military relation theories, adapted to reflect Ghana's unique cultural and institutional context.

As previously stated in *Norman's Identity Politics in Ghana (2023),* "when it comes to competitive, higher order freedoms and ontological security, such as freedom from being denied promotion, freedom from participating in national decision-making, freedom from political and social exclusion, the situation is discouraging." These inequalities frequently manifest as identity-based barriers, including sex-based disadvantages that shape whether a person is employed, promoted, assigned meaningful duties, or recognised within the institution [6].

Ghana's political economy and the broader security services environment also exhibit systemic discrimination based on tribe, religion, and gender. This pattern is evident across public and private institutions, including universities and agencies under the Public Service Commission. Yet, directors, or commanders, and senior administrators often deny or minimise the existence of discriminatory hiring and management practices, despite substantial evidence to the contrary [6].

Throwing the complex and contentious mix of affirmative action programmes into Ghana's already divisive and hierarchical military culture is likely to exacerbate existing tensions within the institution. The introduction of such policies may deepen the social and professional distance between men and women, amplify longstanding disparities, and further entrench the control exercised by the military elite over a personnel structure already marked by fragmentation and uneven opportunities.

## 1.4. Opposition and movements of Affirmative Action programmes

Many researchers and policymakers have presented opposing views on affirmative action programmes in Ghana and elsewhere, resulting in mixed interpretations and contested outcomes. In Ghana, some critics argued that:

"Promoting affirmative action solely for the benefit of women in African societies such as Ghana constitutes a legislative overreach. International supporters of affirmative action implicitly characterise African cultures as inherently hostile and unfit for the flourishing of women; therefore, the government must forcibly intervene to reshape society. However, the critics claim that affirmative action seeks to disrupt what they consider the 'natural' patriarchal order, potentially elevating women at the expense of men and contributing to what they describe as the 'emasculation' of African men" [2].

Similar critiques have emerged in the United States, where some groups portray Diversity, Equality and Inclusivity (DEI) initiatives as attempts to undermine men's economic and social standing. These sentiments have contributed to the growing backlash against DEI efforts and are often cited in discussions surrounding the 2023 U.S. Supreme Court ruling that ended care-conscious admissions policies. In *Students for Fair Admissions (SFFA) Harvard and UNC*, the Court held that race-based admissions violated constitutional principles, effectively dismantling affirmative action policies that had been in place for over two decades and that had historically supported racial minorities and women.

## 1.5. Entry test battery of medical schools

Whereas the GAF and other security agencies rely on physical ability assessments during recruitment, medical schools rely on scholastic and intellectual evaluations to select candidates capable of completing medical training. The training of medical doctors is standardised, objective, and not inherently skewed toward either sex. Once individuals successfully pass the required written and practical examinations, which are based on clearly defined, uniform evaluation criteria, the issue of professional qualifications is no longer in doubt.

Despite this parity in qualification, women remain markedly underrepresented in senior leadership roles within the Ghana Health Service, raising concerns about the consistent appointment of men as Director-General at both the national and regional levels. This pattern suggests the presence of a systemic glass-wall that restricts women's upward mobility in the medical profession and advantages men without a demonstrable merit-based rationale for leadership selection in Ghana's health administration.

## 1.6. Affirmative Action Act as Self-Regulating Law

Although the Affirmative Action Act of 2024 has not yet produced a Standard Operating Procedure (SOP), it has established the programme's foundational structure and outlined the core values expected to guide the forthcoming Legislative Instrument (LI) [2]. The drafting of this LI is critical, as it will operationalise the Act. Unlike an Executive Order (EO), an LI carries greater legal authority in Ghana because of its legislative character and curative capacity under Ghanaian jurisprudence.

Debates surrounding affirmative action in other jurisdictions demonstrate that these policies often undergo cycles of endorsement and opposition. In the United States, public controversy has risen and receded across several decades, with major Supreme Court decisions shaping the landscape. The first wave began in the 1970s, culminating in *Regents*

*of the University of California v. Bakke, 438 U.S. 265 (1978),* in which the Court ruled that rigid racial quotas were unacceptable. Subsequent debates in the 1990s and early 2000s culminated in *Grutter v. Bollinger, 539 U.S. 306 (2003),* which upheld race-conscious admissions as part of holistic review, and *Bostock v. Clayton County, 590 U.S. 664 (2020),* which extended Title VII protections to gender and sexuality.

The most recent shift occurred in 2023, when the Supreme Court, in *Students for Fair Admissions, Inc. (SFFA) v. President and Fellows of Harvard College (Harvard) and SFFA v. University of North Carolina (UNC),* overturned race-conscious admissions policies, signifying a substantial departure from prior rulings. These developments are historically linked to *Brown v. Board of Education, 347 U.S. 483 (1954),* which abolished the "separate but equal" doctrine and broadened educational access for African Americans.

In Ghana, public discussions regarding the Affirmative Action Act have begun, yet it has not yet attained widespread national prominence, mainly due to the implementation mechanisms, particularly the LI. With the commencement of the operational phase, similar patterns of discourse and contestation may arise, influenced by Ghana's unique legal, cultural, and institutional context.

## 2. Methods

The study assessed the first schedule of the Affirmative Action Act of 2024 (Act 1121) to determine compliance obligations for public service organisations and security services under Sections 15(1) – (3) and 16(1) – (5), which mandate *"Progressive Achievements from 2024 to 2034."* Act 1121 requires that between 2024 and 2026, at least 30% of personnel at all levels be women, increasing to 35% between 2027 and 2028, and 50% between 2029 and 2034. Professional progression in the public service and security services, using the Ghana Police Service (GPS) as the primary case study, is examined to assess whether there is historical or current prima facie compliance with these statutory targets.

The study employs a quantitative, cross-sectional design to explore the relationship between sex and access to senior leadership positions in Ghana's public sector, with a focus on the GPS. The GPS is used as a representative case of broader patterns across Ghana's security services and public administration. By examining rank distributions, the study evaluates whether an officer's sex is associated with their position in the rank hierarchy, specifically whether female officers are disproportionately concentrated in lower ranks compared to male officers.

A total of 43,968 valid officer records were analysed, each containing information on sex (male or female) and rank group. For analytical clarity, the detailed police ranks were aggregated into three ordered rank categories, as given in Table 1. These three rank groups [lower, middle (Inspectorate), and senior ranks] represent a progression in authority and responsibility within the police hierarchy and were treated as categorical variables in the analysis. The dataset was obtained from the GPS through the Institute of Security, Disaster and Emergency Studies (ISDES), which granted ethical approval (Ref: ISDES/ Vol. 0004/No. 5/GHSS-16/3/2024/isdes/gh). The data was formally released to the research team on 28/04/2025 for academic use and publication.

A chi-squared test of independence was first conducted at the 0.05 significance level to determine whether the rank group distribution differed by sex. A contingency table (Table 2) was constructed to show the distribution of males and females across the three rank groups. All assumptions for the chi-square test were met, with no cells exhibiting expected

**Table 1. Rank groups and their corresponding police ranks in the GPS.**

| Rank Group | Included Ranks |
|---|---|
| Other Ranks | Constable, Lance Corporal, Corporal, Sergeant |
| Inspectorate | Inspector, Chief Inspector |
| Senior Police Officers (SPOs) | Assistant Superintendent of Police (ASP), Deputy Superintendent, Superintendent, Chief Superintendent, Assistant Commissioner of Police, Deputy Commissioner, Commissioner of Police |

**Table 2. Distribution of male and female officers across rank groups.**

| Group Ranks | | Sex | | Total |
|---|---|---|---|---|
| | | Female | Male | |
| Lower Ranks | Count | 9113 | 19475 | 28588 |
| | Expected Count | 8416.8 | 20171.2 | 28588 |
| Inspectorate | Count | 3327 | 9864 | 13191 |
| | Expected Count | 3883.7 | 9307.3 | 13191 |
| SPO | Count | 505 | 1684 | 2189 |
| | Expected Count | 644.5 | 1544.5 | 2189 |
| Total | Count | 12945 | 31023 | 43968 |
| | Expected Count | 12945 | 31023 | 43968 |

counts below 5. Symmetric effect-size statistics, including Cramer's V and Phi, were used to assess the strength of association. The analysis was performed using SPSS Version 23.

In addition to the chi-square test, an ordinal logistic regression (OLR) model was estimated to examine the effect of sex on an officer's likelihood of occupying higher ranks. Ordinal logistic regression is appropriate when the response variable is binary (dichotomous), as in this study. To estimate the model's parameters, a connecting function is required. In particular, ordinal logistic regression is characterised by response variables and predictors that have an ordinal scale. Let the response variable $Y$ contain $k$ categories, and let $x_i = x_{i1}, x_{i2}, x_{i3}, \ldots x_{iq}$ represent a vector of predictor variables, $q$, at the value of the first observation ($i = 1, 2, \ldots, m$). Opportunities for the $k$-response variable and predictor $X$ are given as $P(Y \leq k \mid x)$ (see [19]). The cumulative opportunities are described as follows: [20]

$$P(Y \leq k \mid x) = \frac{\exp\left(\beta_{0k} + \sum_{r=1}^{q} \beta_r x_{ir}\right)}{1 + \exp\left(\beta_{0k} + \sum_{r=1}^{q} \beta_r x_{ir}\right)}.$$

The following formula expresses the cumulative logit model,

$$\text{logit}\left[P(Y \leq k \mid x)\right] = \log\left(\frac{P(Y \leq k \mid x)}{P(Y > k \mid x)}\right)$$

$$= \log\left(\frac{P(Y \leq k \mid x)}{1 - P(Y \leq k \mid x)}\right)$$

$$= \beta_{0k} + \sum_{r=1}^{q} \beta_r x_{ir}$$

where $k = 1, 2, \ldots, K-1$, and $\beta_{0k}$, $\beta_r$ are the vector regression coefficients.

The study employed a logit link function, which adheres to the proportionate odds assumption. The rank group is coded as 1 = lower ranks, 2 = inspectorate, and 3 = senior police officers. The independent variable, sex, was coded as 0 for female and 1 for male, with male officers treated as the reference category.

Null Hypotheses

   (HO₁): There is no association between an officer's sex and their rank group in the Ghana Police Service.

   (HO₂): An officer's sex does not significantly predict their likelihood of occupying a higher or lower rank.

Alternative Hypotheses

   (HA₁): There is a significant association between an officer's sex and their rank group.

   (HA₂): An officer's sex significantly predicts their likelihood of occupying a higher or lower rank.

## 3. Results and discussion

The results section is divided into four parts, with the first summarising the findings from the literature review and the legal framework review. Two hypotheses are tested, and the findings are reported.

### 3.1. Expected deliverables under the Affirmative Action Act of 2024

The Affirmative Action Act of 2024 can be interpreted as largely self-executing due to the activities listed in schedules one to seven. These schedules cover: (1) processes for measuring progressive compliance with targets; (2) international conventions; (3) guidelines and strategies for gender equity in relation to the public service; (4) strategies for gender equity in relation to the executive; (5) strategies for gender equity in relation to the judiciary, (6) strategies for gender equity in relation to parliament and the parliamentary service; and (7) strategies for gender equity in relation to political parties affixed to the Act. In addition to these specific schedules, the Act contains directive principles, under Sections 1 and 2 ("Object of the Act"), which impose a compliance obligation on designated institutions, including Section 16 (1)-(5) on gender equity in the security services.

The enforceability of these provisions, however, raises practical concerns. While the Act seeks to accelerate representational equity, its directive principles may inadvertently put pressure on command-based structures, such as the security services, particularly in relation to training standards, recruitment benchmarks, and rank progression criteria. This challenge becomes more apparent when compared with the quantitative expectations stated in the First Schedule, which mandates a system for measuring progressive compliance with representational targets.

The First Schedule further requires institutions to provide baseline data for the year 2018 (First Schedule, 1(a), p. 20). However, selecting 2018 as the baseline year lacks empirical justification. The selection was not based on any empirical data to assess the appropriateness of the 2018, and society is being redesigned from scratch without knowing the aetiology, antecedents, or demographic facts about the public challenge. As such, the Act appears to assume a neutral starting point, although institutional and systemic discrimination, especially gender-based discrimination, is historically embedded rather than emerging in 2018 [21]. This raises questions about the methodological soundness and historical sensitivity of the selection of the baseline year.

The First Schedule (2) additionally obliges institutions to submit annual compliance reports, after which the committee may summon relevant officials to clarify issues in the report, make appropriate recommendations, and provide a compliance certificate. On these bases, Act 1121 of 2024 may be described as self-executing.

Nonetheless, characterising the Act as entirely self-executing overlooks Ghana's legislative procedures, particularly the requirement for subsidiary legislation (Legislative Instrument) to operationalise statutory provisions. Without a supporting Legislative Instrument, implementation, monitoring, sanctions, and reporting protocols may lack procedural clarity and enforceable regulatory detail.

### 3.2. Systematic advancement of male doctors over their female counterparts

Patterns in progression within Ghana's public health institutions reveal a persistent gender imbalance in senior leadership and career advancement. While security services have typically been foregrounded in discussions of gender disparities, similar institutional patterns are evident within the Ministry of Health's major agencies, such as the Ghana Health Service (GHS) and the National Health Insurance Scheme (NHIS). Since the commencement of the 4th republican governance system in 1992, all Director-Generals of the GHS have been male, except for the 2023/2024 appointment cycle, indicating a longstanding gender representation gap. The distribution of the Regional Health Directors and other key leadership positions, as presented in Table 3, further illustrates the concentration of senior administrative and medical authority in male hands.

**Table 3. Distribution of Regional and Divisional Directors in the GHS by Sex.**

| Role Category | Male | Female | Total |
|---|---|---|---|
| Regional Directors | 9 | 1 | 10 |
| Divisional Heads | 8 | 2 | 10 |
| **Total** | **17** | **3** | **20** |

Source: Divisional Directorate, Moh.Gov.gh, 2025, ghs.gov.gh, 2024.

### 3.3. Systematic discrimination against women in the National Health Insurance Scheme

Within the NHIS, women's representation in upper management remains critically limited. Of the 16 regional directorates within NHIS, only one is headed by a woman, and among the 275 district directorates, only three are led by women. At the NHIS headquarters, fewer than 5 women hold director-level positions, and the number of female deputy directors is similarly low. This pattern highlights a structural ceiling effect in which women's advancement remains markedly restricted despite equivalent professional qualifications and service tenure. An interesting observation is that directorship positions at both national and regional levels have not only been predominantly occupied by men, but some of these male personalities possess less than optimal qualifications. This disparity underscores the need for a systematic review of internal promotion procedures to ensure transparent merit-based criteria. Accordingly, the current study is positioned to explore the determinants of these inequalities and to suggest remedial actions to accelerate corrective measures to mitigate the negative social impact on the victims of the Affirmative Action Act in Ghana.

### 3.4. Auxiliary findings in the security agencies

Patterns of leadership appointments in the security agencies point to a complicated mix of institutional, socio-political, and identity-based considerations. While formal promotion procedures exist, post-retirement contract extensions within the higher echelons of the services are often influenced by perceived political alignment, further shaped by factors such as ethnicity, religious affiliation, and neo-patrimonial networks. Such appointments sometimes show little regard for qualifications, gender, capability or the necessity of the role. This emphasises the necessity for clear and uniform promotion systems, especially in structured services where hierarchical legitimacy and command authority are essential to institutional effectiveness.

### 3.5. One female Inspector-General of Police since 1969

As of March 2025, the Ghana Police Service comprised 43,968 officers, of whom 12,945 (30%) were female, and 31,023 (70%) were male. Despite the steady increase in female participation in the service, gender disparities persist at the highest levels of leadership. Since Ghana assumed full control of its police service in 1969, only one woman has served as Inspector-General of Police (IGP), and even then, only in an acting capacity. Table 4 presents the chronological list of individuals who have served as IGP and illustrates the continued concentration of executive police leadership among male officers.

### 3.6. Three female prison Director-Generals since 1963

Ghana Prison Service in April of 2025, got its third female Director-General since independence with the appointment of Patience Baffoe-Bonnie. The Second female Director-General, Matilda Baffour Awuah, served in 2013, while the first, Victoria Zormelo-Gorleku, was appointed in 1963 [22]. Apart from these female appointments across more than six decades, all other occupants of the office have been male. This ongoing trend illustrates the limited representation of women in executive leadership positions.

**Table 4. Appointment History of Inspectors-General of Police in Ghana, 1969–2025.**

| No. | Name | Tenure | Gender |
|---|---|---|---|
| 1 | Bawa Andani Yakubu | 1969–1971 | Male |
| 2 | R. D. Ampaw | 1971–1972 | Male |
| 3 | J. H. Cobbina | 1972–1974 | Male |
| 4 | Ernest Ako | 1974–1978 | Male |
| 5 | B. S. K. Kwakye | 1978–1979 | Male |
| 6 | C. O. Lamptey | Jun–Sep 1979 | Male |
| 7 | F. P. Kyei | 1979–1981 | Male |
| 8 | R. K. Kuglenu | 1981–1984 | Male |
| 9 | S. S. Omane | 1984–1986 | Male |
| 10 | C. K. Dewornu | 1986–1989 | Male |
| 11 | J. Y. A. Kwofie | 1990–1996 | Male |
| 12 | Peter Nanfuri | 1996–2001 | Male |
| 13 | Ernest Owusu-Poku | Jan–Jul 2001 | Male |
| 14 | Nana Owusu-Nsiah | 2001–2005 | Male |
| 15 | Patrick K. Acheampong | 2005–2009 | Male |
| 16 | Elizabeth Mills-Robertson | Jan–May 2009 (Acting) | Female |
| 17 | Paul Tawiah Quaye | 2009–2013 | Male |
| 18 | Mohammed A. Alhassan | 2013–2015 | Male |
| 19 | John Kudalor | 2016–2017 | Male |
| 20 | David Asante-Apeatu | 2017–2019 | Male |
| 21 | James Oppong-Boanuh | 2019–2021 | Male |
| 22 | George Akuffo-Dampare | Aug 2021–Mar 2025 | Male |
| 23 | Christian Tetteh Yohunu | 2025–Present | Male |

Source: IGPs compilation based on official Government of Ghana records and Ghana Police Service publications.

### 3.7. Empirical assessment of gender representation in police ranks

The chi-square test revealed a significant relationship between sex and rank group within the GPS. The Pearson chi-square value was 237.478, with a p-value less than 0.05, indicating that the null hypothesis of independence between sex and rank can be rejected (see Table 5). This confirms that rank distribution varies systematically by sex rather than occurring by chance.

   Descriptive statistics provided further insight into the characteristics of the relationship between sex and rank. Female officers constituted 31.9% of the lower ranks, but their representation declined to 25.2% at the inspectorate level and dropped further to 23.1% among SPOs. Conversely, the proportion of male officers increased steadily across the hierarchy, reaching 68.1% in the lower ranks, 74.8% in the inspectorate, and 76.9% among SPOs. These patterns demonstrate a clear gender imbalance, with women disproportionately concentrated at the lower ranks and markedly underrepresented at the higher leadership levels. This distribution aligns with the chi-square test results and suggests persistent structural barriers that limit women's upward mobility within the GPS.

   Cramer's V was calculated to assess the strength of the association, yielding a value of 0.073 (p = 0.000). Although statistically significant, this value indicates a weak association between sex and rank group. In practical terms, this suggests that while gender appears to influence rank distribution, the magnitude of this effect is relatively small. Such outcomes are common in large administrative datasets, where even modest differences can achieve statistical significance.

**Table 5. Results of the Chi-Square Test of Independence between Sex and Rank Group in the Ghana Police Service.**

*Chi-Square Tests*

| | Value | Df | Asymptotic Significance (2-sided) | |
|---|---|---|---|---|
| Pearson Chi-Square | 237.478[a] | 2 | 0.00 | |
| Likelihood Ratio | 241.787 | 2 | 0.00 | |

*Symmetric Measures*

| | Value | Approximate Significance | | |
|---|---|---|---|---|
| Nominal by Nominal | | | | |
| Phi | 0.073 | 0.00 | | |
| Cramer's V | 0.073 | 0.00 | | |
| N of Valid Cases | | 43968 | | |

Although the chi-square test does not establish causality, the significant association suggests that structural or institutional factors may be contributing to gender disparities in rank attainment. These factors may include unequal access to career development opportunities, disparities in mentorship, biases in promotion criteria, or historical recruitment patterns. The modest effect size does not diminish the importance of addressing the issue, especially in the context of institutional equity, representation, and compliance with Affirmative Action requirements.

The ordinal logistic regression(OLR) analysis, as shown in Table 6, revealed that sex was a statistically significant predictor of rank group. The odds ratio (Exp($\beta$) ≈ 0.709) indicates that female officers are approximately 29% less likely than their male counterparts to occupy higher ranks (Inspectorate or Senior Police Officers), holding all else constant. The 95% confidence interval for the coefficient (–0.388 to –0.300) further confirms the precision and robustness of this effect.

The goodness-of-fit statistics for the ordinal logistic regression model indicate that the model fits the data well. According to the results in Table 7, both the Pearson and deviance chi-square tests yielded non-significant results (Pearson $\chi^2 = 0.003$, $p = 0.957$; deviance $\chi^2 = 0.003$, $p = 0.957$), indicating no meaningful difference between the observed and model-predicted values and confirming acceptable model fit. However, the pseudo R-squared values were relatively low (Cox & Snell $R^2 = 0.005$, Nagelkerke $R^2 = 0.007$, and McFadden $R^2 = 0.003$), suggesting that although sex is a statistically significant predictor of police rank, it accounts for less than 1% of the variance in rank outcomes. This implies that other unmeasured factors likely play a more substantial role in determining officers' rank progression.

Further analysis was conducted to determine whether sex is associated with appointment to senior leadership roles within Ghana's public service, specifically comparing the Ghana Health Service (GHS) and the GPS. A Chi-square test of independence was performed on a two-way contingency table consisting of 2,209 individuals: 20 regional directors/divisional heads within the GHS, and 2,189 SPOs in the GPS. Of the total, 1,701 were male and 508 were female.

The results showed a statistically significant association between sex and leadership role category, $\chi^2$ (1, N = 2209) = 4.48, $p = 0.034$. Although men dominate both leadership categories, women are proportionally more represented among

**Table 6. Ordinal Logistic Regression Coefficients and Odds Ratios Predicting Police Rank Category from Officer Sex.**

| | | Estimate | Std. Error | Wald | df | Sig. | 95% Confidence Interval | |
|---|---|---|---|---|---|---|---|---|
| | | | | | | | Lower Bound | Upper Bound |
| Threshold | [Rank_Order = 1] | 0.523 | 0.012 | 1992.265 | 1 | 0 | 0.5 | 0.546 |
| | [Rank_Order = 2] | 2.858 | 0.023 | 15972.435 | 1 | 0 | 2.814 | 2.903 |
| Location | [Sex_of_officers=0] | −0.344 | 0.022 | 236.436 | 1 | 0 | −0.388 | −0.3 |
| | [Sex_of_officers=1] | 0[a] | . | . | 0 | . | . | . |

[a]This parameter is set to zero because it is redundant.

**Table 7. Chi-Square Test of Independence Comparing Gender Representation in GHS Leadership and GPS Senior Police Officers.**

| Goodness-of-Fit | | | |
|---|---|---|---|
| | Chi-Square | df | Sig. |
| Pearson | 0.003 | 1 | 0.957 |
| Deviance | 0.003 | 1 | 0.957 |
| *Pseudo R-Square* | | | |
| Cox and Snell | 0.005 | | |
| Nagelkerke | 0.007 | | |
| McFadden | 0.003 | | |

SPOs than among top-tier GHS directors. Only 3 of the 20 GHS regional directors and divisional heads were female (15%), compared to 505 of the 2,189 SPOs in the GPS (23%). These numbers suggest that women are markedly under-represented in the most elite leadership appointments within the health sector, even relative to their already limited representation in senior policing roles, underscoring pervasive gender disparities in Ghana's public sector leadership landscape.

## 4. Concluding remarks

Both the chi-square test and the ordinal logistic regression analysis provide strong statistical evidence that rank distribution in the GPS is associated with sex. The chi-square results indicate a pattern of declining female representation with rising rank, while the regression model confirms that female officers are statistically less likely than male officers to attain higher-ranking positions.

Although the effect is statistically significant, the small value of Cramer's V (0.073) and low pseudo R-squared values (< 0.01) suggest that sex alone is not a strong determinant of rank outcomes. This result underscores the need to consider additional explanatory factors such as years of service, professional development, regional deployment, and institutional promotion policies.

These findings point to a persistent gender disparity in police leadership. While more women are entering the police service, their progression into senior roles remains disproportionately low. Addressing this imbalance will require targeted reforms, including gender-sensitive promotion criteria, equitable access to specialised and leadership training, and structured mentorship opportunities for female officers.

In conclusion, sex significantly influences rank placement within the Ghana Police Service; however, additional determinants of career progression must be explored. Strengthening gender equity is essential to building a diverse and inclusive leadership structure in law enforcement.

### 4.1. Are women inherently unfit for high-level security or health leadership positions?

The empirical findings provide no basis for any claim that women are not "combat-ready," incapable of critical decision-making, or physically unsuited for high-level roles such as Director-General of Health, Inspector-General of Police, or Brigadier-General. Between 1961 and 2024, the leadership of the GAF was exclusively male, reflecting entrenched paternalistic and exclusionary practices inherited from colonial military structures. However, this historical pattern does not indicate female incapacity; rather, it reflects institutional design, gendered norms, and role segregation, not merit.

Globally, the traditional military model restricted women to support roles rather than combat roles. Over time, this restriction eased, and many countries now allow women to enter a wide range of combat and technical units. Combat exposure provides critical operational competencies such as decision-making, logistics, and command experience, which

are often prerequisites for promotion. Historically, Ghanaian women were confined mainly to clerical or administrative roles, limiting opportunities to acquire such expertise [4].

The administration of power was based on paternalistic and segregated standards, similar to patterns seen in the colonial-era security organisation in Ghana, Nigeria, and South Africa under the apartheid regime between 1948 and 1994 [5,6]. Before the promotions of Brigadier-General Edjeani-Afenu and Brigadier General Twum-Barima, the GAF functioned as a male-dominated military establishment, shaped by unintended discrimination, gender segregation, and institutional directives that reinforced male control. Civilian administrations also appear to have supported such a situation, particularly the executive leadership, at the expense of women's interests and status in the military.

Military institutions around the world have historically relied on uniform physical test batteries to assess recruit suitability and determine officers' career trajectories. These tests have often been gender-neutral in design but gender-biased in effect, disadvantaging women and reinforcing perceptions of inferior combat readiness.

During the 2023 International Day for Women events in Accra, the GAF announced that it had met and exceeded UN-recommended quotas for female inclusion, noting that women were now being posted to combat units and allowed to take several formerly male-dominated courses [23]. However, anecdotal progress does not necessarily reflect advancement to senior ranks. To evaluate the impact of the developments for females in the army, it would have been more useful to examine the challenges posed by the Occupational Physical Assessment Test (OPAT) and Military Occupational Specialties (MOS).

Promotion in the military, both in Ghana and elsewhere, depends on MOS pathways, which are shaped by initial physical test performance. Without analysing the comparative progression rates of men and women to senior ranks such as Colonel, Brigadier-Generals, and Major-Generals, it is difficult to assess the substantive impact of recent reforms.

Baldwin and Rothwell's [24] long-term evaluation of 160,000 U.S. Air Force promotion cases found that women were severely underrepresented in officer corps positions and that white males were promoted faster than white females. Although women sometimes had competitive promotion rates, structural inequalities persisted across ranks.

Similarly, research shows that women are often perceived as less combat-ready, even when objective evidence is lacking [25]. This perception has been used to justify restricting women's promotions in many militaries.

Ghana, unlike nations engaged in continuous warfare, rarely deploys soldiers into active combat. A Ghanaian soldier can complete an entire career, even up to the rank of General, without experiencing war, relying instead on peacekeeping missions that focus on security enforcement, negotiation, and humanitarian activities. Thus, arguments invoking "combat readiness" as a barrier to women's promotion are not well grounded in Ghana's operational reality.

The argument in favour of combat readiness as a demerit against the promotion of women in the military to higher ranks in some African nations, has largely been based on physical test batteries which are not gender-integrated and ultimately affect military operating procedures and role assignments [26].

Foulis et al. [27] examined the U.S. Army's creation of a scientifically validated screening instrument for combat arms MOS. A substantial, mixed-gender cohort (608 males, 230 females) indicated that four principal tests, medicine ball throw, squat lift, beep test, and standing long jump, were predictive of combat performance. The OPAT was therefore designed as a test battery that doesn't favour any one gender or age group and assesses physical ability [25,28,29]. Israel, one of the few countries that requires women to serve, also found clear differences in how well men and women performed on physical tests. These were attributed not only to intrinsic anthropometric variations but also to pre-recruitment lifestyle disparities [26]. Even so, Israel maintains one of the world's most gender-integrated armed forces.

### 4.2. Recommendations

***The Commanding Officers of Security Agencies*** It is recommended that the Command Leadership undertake an internal review of gender-representation practices and establish formal redress mechanisms to address long-standing disparities in promotion and career progression. This may include structured gender-equity communication, acknowledgement

of advancement barriers, and the development of clear pathways that support women in attaining senior ranks. Such measures should reinforce institutional confidence, strengthen morale, and align advancement procedures with statutory expectations under the Affirmative Action Act.

*The Appointing Authority* It is further recommended that the appointing authorities consider merit-based recalibration of senior leadership positions in accordance with Sections 15 and 16 of the Affirmative Action Act of 2024. This may involve reviewing recent appointments to ensure they align with representational and competency benchmarks, and ensuring that qualified female officers are equitably considered for executive and command roles. This process should remain grounded in transparent criteria, meritocracy, and institutional stability, while advancing compliance with the Act's gender-equity provisions.

## Acknowledgments

The authors gratefully acknowledge the Institute for Security, Disaster and Emergency Studies, Langma, Ghana, for providing access to the administrative rank data used in this study. The authors also thank the Ethics Committee of the Institute for Security, Disaster and Emergency Studies for its careful review and ethical oversight of the research.

## Author contributions

**Conceptualization:** Ishmael D. Norman, Emmanuel Dodzi Kpeglo, Saralees Nadarajah.

**Data curation:** Emmanuel Dodzi Kpeglo, Saralees Nadarajah.

**Formal analysis:** Emmanuel Dodzi Kpeglo, Saralees Nadarajah.

**Funding acquisition:** Ishmael D. Norman, Saralees Nadarajah.

**Investigation:** Ishmael D. Norman.

**Methodology:** Ishmael D. Norman, Emmanuel Dodzi Kpeglo, Saralees Nadarajah.

**Project administration:** Ishmael D. Norman.

**Resources:** Ishmael D. Norman, Saralees Nadarajah.

**Software:** Emmanuel Dodzi Kpeglo, Saralees Nadarajah.

**Supervision:** Ishmael D. Norman, Saralees Nadarajah.

**Validation:** Emmanuel Dodzi Kpeglo.

**Visualization:** Ishmael D. Norman, Emmanuel Dodzi Kpeglo.

**Writing – original draft:** Ishmael D. Norman, Emmanuel Dodzi Kpeglo, Saralees Nadarajah.

**Writing – review & editing:** Ishmael D. Norman, Emmanuel Dodzi Kpeglo, Saralees Nadarajah.

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
