## [Decision Letter · Decision Letter 0]

30 Oct 2025

PONE-D-25-46545Sex and Rank in Public Service Hierarchies: Rank Distribution in the Ghana Police ServicePLOS ONE

Dear Dr. Kpeglo,

Thank you for submitting your manuscript to PLOS ONE. After careful consideration, we feel that it has merit but does not fully meet PLOS ONE’s publication criteria as it currently stands. Therefore, we invite you to submit a revised version of the manuscript that addresses the points raised during the review process.

 Your manuscript has been evaluated by a reviewer, and their comments are available below. They request improvements to the reporting of the sampling process and statistical analysis. Please carefully revise your manuscript to address the points raised. Please note that we have only been able to secure a single reviewer to assess your manuscript. We are issuing a decision on your manuscript at this point to prevent further delays in the evaluation of your manuscript. Please be aware that the editor who handles your revised manuscript might find it necessary to invite additional reviewers to assess this work once the revised manuscript is submitted. However, we will aim to proceed on the basis of this single review if possible. 

We look forward to receiving your revised manuscript.

Kind regards,

Jenna Scaramanga

Staff Editor

PLOS ONE

3. In the online submission form you indicate that your data is not available for proprietary reasons and have provided a contact point for accessing this data. Please note that your current contact point is a co-author on this manuscript. According to our Data Policy, the contact point must not be an author on the manuscript and must be an institutional contact, ideally not an individual. Please revise your data statement to a non-author institutional point of contact, such as a data access or ethics committee, and send this to us via return email. Please also include contact information for the third party organization, and please include the full citation of where the data can be found.

Reviewers' comments:

Reviewer's Responses to Questions

**Comments to the Author**

1. Is the manuscript technically sound, and do the data support the conclusions?

Reviewer #1: Yes

2. Has the statistical analysis been performed appropriately and rigorously? 

Reviewer #1: Yes

3. Have the authors made all data underlying the findings in their manuscript fully available?

Reviewer #1: No

4. Is the manuscript presented in an intelligible fashion and written in standard English?

Reviewer #1: Yes

5. Review Comments to the Author

Reviewer #1: This manuscript examines gender disparities in leadership across Ghana’s security and health institutions, focusing empirically on the Ghana Police Service (GPS). Using administrative records (N=43,968), you show that women are underrepresented at higher ranks and document a statistically significant association between sex and rank (χ²), corroborated by ordinal logistic regression (OR≈0.71). The paper’s interdisciplinary framing—civil–military relations theory and the Affirmative Action Act (2024)—adds policy salience and broader relevance. Replace “available on request” with public deposition of an anonymized dataset sufficient to reproduce Tables 2, 5–7 and the OLR, along with SPSS syntax/output. If restrictions exist, specify the legal/ethical constraints, provide a minimal dataset, and indicate a data access mechanism consistent with PLOS policy. The text references 55,129 GPS officers (as of March 2025) but the analytic sample is 43,968 valid records.

Please reconcile: e.g., explain missingness, exclusions (invalid/duplicate/missing rank), or time windows.Condense extended discussions of U.S. affirmative action jurisprudence; ensure precision (e.g., Bostock v. Clayton County concerns Title VII sex discrimination, not admissions-based affirmative action).

Move lengthy quotations and some theoretical background to Supplementary Material, summarizing key points in the main text.

Standardize terminology and ensure consistent capitalization. Ensure table labelling is self-contained. Proofread for redundancies and minor grammatical issues; trim long sentences for readability.

This is a promising, policy-relevant manuscript. With data-sharing compliance, clarified sampling, strengthened statistical reporting, and a more concise, neutral presentation, the paper would make a solid contribution to the literature on gender and public-sector hierarchies in Ghana.

6. PLOS authors have the option to publish the peer review history of their article (what does this mean? ). If published, this will include your full peer review and any attached files.

**Do you want your identity to be public for this peer review?** For information about this choice, including consent withdrawal, please see our Privacy Policy .

Reviewer #1: No

---

## [Author Response · Author response to Decision Letter 1]

17 Dec 2025

RESPONSE TO EDITORIAL TEAM COMMENTS

Response to Comment 1

Thank you for the feedback. I confirm that this study involved the analysis of fully anonymised secondary administrative data from the Ghana Police Service. No identifiable personal information (names, service numbers, contact details, or any traceable identifiers) was accessed by the research team at any stage.

As indicated in the Ethical Clearance Letter and Data Access Letter (both already uploaded to the system):

1. The Institute for Security, Disaster and Emergency Studies (ISDES) granted ethical approval for the study.

2. The data were provided by ISDES after they had conducted the review and anonymisation.

3. No medical records or patient samples were used.

Response to Comment 2

Thank you for alerting me. I have corrected the repository link in the Data Availability Statement. The dataset has now been re-uploaded. The updated link is: https://zenodo.org/records/17747718

RESPONSE TO REVIEWER COMMENTS

Comment 1

Replace “available on request” with public deposition of an anonymized dataset sufficient to reproduce Tables 2, 5–7 and the OLR, along with SPSS syntax/output. If restrictions exist, specify the legal/ethical constraints, provide a minimal dataset, and indicate a data access mechanism consistent with PLOS policy.

Response

Thank you for this important recommendation. An anonymised minimal dataset sufficient to reproduce all statistical analyses reported in this study (including Tables 2 and 5–7) has been deposited in a public repository (https://zenodo.org/records/17747718). The repository also contains the SPSS syntax file (analysis_syntax.sps) and the SPSS output file (analysis_output.spv) used to conduct the chi-square tests and the ordinal logistic regression.

Comment 2

“The text references 55,129 GPS officers (as of March 2025) but the analytic sample is 43,968 valid records. Please reconcile: e.g., explain missingness, exclusions (invalid/duplicate/missing rank), or time windows.”

Response

Thank you for drawing attention to this inconsistency. The manuscript has now been clarified. The correct number of serving Ghana Police Service personnel as of March 2025 is 43,968, comprising 12,945 females (30%) and 31,023 males (70%). This represents the whole administrative population extracted from the Personnel Management Information System for the analysis. The figure of 55,129 does not apply to the period under study, and any possible ambiguity has now been removed from the text.

Comment 3

“Condense extended discussions of U.S. affirmative action jurisprudence; ensure precision (e.g., Bostock v. Clayton County concerns Title VII sex discrimination, not admissions-based affirmative action).”

Response

Thank you for this helpful observation. We have substantially condensed the discussion of U.S. affirmative action jurisprudence to maintain focus on the Ghanaian context. The revised text now briefly references key U.S. cases only to illustrate the general evolution of equality jurisprudence. We also corrected the earlier imprecision regarding Bostock v. Clayton County and now clarify that it concerns Title VII sex discrimination rather than admissions-based affirmative action. Longer quotations and extended legal background have been moved to the Supplementary Material (Section A1). These updates improve accuracy, conciseness, and alignment with the objectives of the paper.

Comment 4

“Move lengthy quotations and some theoretical background to Supplementary Material, summarizing key points in the main text.”

Response

Thank you for this valuable recommendation. We have now streamlined the main text by removing lengthy quotations and extended theoretical discussions. In the manuscript, we substituted concise summaries that preserve the key conceptual and legal arguments relevant to the study. This revision improves clarity, readability, and alignment with PLOS formatting expectations.

Comment 5

“Standardize terminology and ensure consistent capitalization.”

Response

Thank you for highlighting this issue. We conducted a comprehensive review of terminology, capitalization, and rank titles across the manuscript. Institutional names, rank designations, and statistical terms have now been standardized.

Comment 6

“Ensure table labelling is self-contained.”

Response

Thank you for this helpful observation. All tables have been revised to ensure that they are fully self-contained.

Comment 7

“Proofread for redundancies and minor grammatical issues; trim long sentences for readability.”

Response

Thank you for this comment. We have conducted a thorough proofreading of the manuscript and made multiple revisions to improve clarity and readability. Redundant phrases were removed, minor grammatical errors corrected, and several long or complex sentences were shortened. These changes enhance the overall flow of the paper without altering its substantive content.

---

## [Decision Letter · Decision Letter 1]

12 Jan 2026

PONE-D-25-46545R1Sex and Rank in Public Service Hierarchies: Rank Distribution in the Ghana Police ServicePLOS One

Dear Dr. Kpeglo,

Thank you for submitting your manuscript to PLOS ONE. After careful consideration, we feel that it has merit but does not fully meet PLOS ONE’s publication criteria as it currently stands. Therefore, we invite you to submit a revised version of the manuscript that addresses the points raised during the review process. Please submit your revised manuscript by Feb 26 2026 11:59PM. If you will need more time than this to complete your revisions, please reply to this message or contact the journal office at plosone@plos.org . Please include the following items when submitting your revised manuscript:

We look forward to receiving your revised manuscript.

Kind regards,

Adetayo Olorunlana, Ph.D.

Academic Editor

PLOS One

Journal Requirements:

Additional Editor Comments:

The topic focuses narrowly on sex and rank distribution within the Ghana Police Service, whereas the abstract extends beyond this scope to include the Ghana Health Service, the Armed Forces, physical assessment tests, and the Affirmative Action Act (2024). This creates a misalignment between the title and the content described in the abstract. Kindly adjust either the topic or the abstract.  A study of this nature should also have acknowledgement.

Reviewers' comments:

Reviewer's Responses to Questions

**Comments to the Author**

1. If the authors have adequately addressed your comments raised in a previous round of review and you feel that this manuscript is now acceptable for publication, you may indicate that here to bypass the “Comments to the Author” section, enter your conflict of interest statement in the “Confidential to Editor” section, and submit your "Accept" recommendation.

Reviewer #1: All comments have been addressed

2. Is the manuscript technically sound, and do the data support the conclusions?

Reviewer #1: Yes

3. Has the statistical analysis been performed appropriately and rigorously? 

Reviewer #1: Yes

4. Have the authors made all data underlying the findings in their manuscript fully available?

Reviewer #1: Yes

5. Is the manuscript presented in an intelligible fashion and written in standard English?

Reviewer #1: Yes

6. Review Comments to the Author

Reviewer #1: All comments have been fully addressed in the revised version of the manuscript. The authors have carefully considered the suggestions raised in the previous review round and incorporated them appropriately. The manuscript is now clear, methodologically sound, and suitable for publication in its present form.

7. PLOS authors have the option to publish the peer review history of their article (what does this mean? ). If published, this will include your full peer review and any attached files.

**Do you want your identity to be public for this peer review?** For information about this choice, including consent withdrawal, please see our Privacy Policy .

Reviewer #1: **Yes: ** Veljko Turanjanin

---

## [Author Response · Author response to Decision Letter 2]

13 Jan 2026

RESPONSE TO EDITOR COMMENTS

Editor Comments:

The topic focuses narrowly on sex and rank distribution within the Ghana Police Service, whereas the abstract extends beyond this scope to include the Ghana Health Service, the Armed Forces, physical assessment tests, and the Affirmative Action Act (2024). This creates a misalignment between the title and the content described in the abstract. Kindly adjust either the topic or the abstract. A study of this nature should also have acknowledgement.

Response

We appreciate this suggestion.

i. In response to the editor’s comment, the title has been revised to “Sex and rank in public service hierarchies: Rank distribution in Ghana’s health and security services” to align fully with the broader institutional scope described in the abstract.

ii. An acknowledgement statement has been added to the revised manuscript, in line with journal requirements, to appropriately recognise institutional support and ethical oversight.

Journal Requirements:

Response

We carefully reviewed the reference list to ensure completeness, accuracy, and consistency with the in-text citations, in line with PLOS ONE requirements. During this review, we identified three references that were included in the reference list but not cited in the main text, namely

1. Encyclopaedia Britannica. (n.d.). Brown v. Board of Education of Topeka. In Britannica.com. Retrieved August 24, 2024, from

https://www.britannica.com/event/Brown-v-Board-of-Education-of-Topeka

2. Loden, M. (1987). Recognising women's potential: No longer business as usual. Management review, 76(12), 44.

3. Norman, I. D. (2024). Identity politics in Sub-Saharan Africa. Cambridge Scholars Publishing. Lady Stephenson Library, Newcastle upon Tyne, NE6 2PA, UK.

These references have been removed from the reference list to ensure that all listed references are explicitly cited in the manuscript.

In addition, the reference list was checked for bibliographic accuracy and relevance. We confirm that no retracted articles are cited in the revised manuscript; therefore, no justification for citing retracted literature or inclusion of retraction notices was required. Any necessary adjustments to the reference list have been made accordingly and are reflected in the revised submission.

---

## [Editor Report · Decision Letter 2]

25 Jan 2026

Sex and rank in public service hierarchies: Rank distribution in Ghana’s health and security services

PONE-D-25-46545R2

Dear Dr. Kpeglo,

We’re pleased to inform you that your manuscript has been judged scientifically suitable for publication and will be formally accepted for publication once it meets all outstanding technical requirements.

Kind regards,

Adetayo Olorunlana, Ph.D.

Academic Editor

PLOS One
---

## [Editor Report · Acceptance letter]

PONE-D-25-46545R2

PLOS One

Dear Dr. Kpeglo,

I'm pleased to inform you that your manuscript has been deemed suitable for publication in PLOS One. Congratulations! Your manuscript is now being handed over to our production team.

Kind regards,

on behalf of

Associate Professor Adetayo Olorunlana

Academic Editor

PLOS One